# Prognostic Significance of Baseline Blood Glucose Levels and Glucose Variability in Severe Acute Kidney Injury: A Secondary Analysis from the RENAL Study

**DOI:** 10.3390/jcm12010015

**Published:** 2022-12-20

**Authors:** Ying Xie, Jin Lin, Martin Gallagher, Rinaldo Bellomo, Xia Wang, Meg Jardine, Meili Duan, Amanda Wang

**Affiliations:** 1The Second Affiliated Hospital, Soochow University, Suzhou 215004, China; 2The George Institute for Global Health, University of New South Wales, Newtown, NSW 2042, Australia; 3Beijing Friendship Hospital, Capital Medical University, Beijing 100052, China; 4Department of Intensive Care, Austin Hospital, Melbourne, VIC 3084, Australia; 5Department of Renal Medicine, Concord General Repatriation Hospital, Concord, NSW 2139, Australia; 6Concord Clinical School, University of Sydney, Concord West, NSW 2138, Australia; 7The Faculty of Medicine and Health Sciences, Macquarie University, Sydney, NSW 2109, Australia

**Keywords:** acute kidney injury, blood glucose levels, glycemic variability, mortality, renal re-placement therapy

## Abstract

Aim: To study the associations between baseline blood glucose levels (BGL), glycemic variability and clinical outcomes in patients with severe acute kidney injury (AKI) receiving continuous renal replacement therapy (CRRT). Methods: We performed a secondary analysis of the Randomized Evaluation of Normal versus Augmented Level of RRT (RENAL) study. A multivariate Cox regression model was used to assess the association between baseline BGL, glycemic variability and clinical outcomes. The primary outcome was all-cause mortality, and secondary outcomes were duration of hospital and intensive care unit (ICU) stay. Results: Baseline BGL data were available in 1404 out of 1508 patients from the RENAL study. Among them, 627 patients died within 90 days of randomization. Compared to patients in the second quartile (BGL 5.8–7.2 mmol/L), patients in the first quartile (BGL < 5.8 mmol/L) had increased mortality rate (90-day HR 1.48; *p* = 0.001; 28-day HR 1.47; *p* = 0.042). However, there were no significant differences in ICU and hospital length of stay (LOS) (*p* = 0.82 and *p* = 0.33, respectively). Glycemic variability data were from 1345 out of 1404 patients who had data for BG values within 28 days. Higher coefficient of variation (CV) (HR 1.02; P trend = 0.002) and standard deviation value (SD) (HR 1.29; P trend = 0.027) were associated with higher risk of death at day 90. Conclusions: We identified a low BGL within the normal physiological range at baseline and greater CV and SD values as significant modifiable risk factors for mortality in severe AKI patients in ICU, which may be a target for intervention.

## 1. Background

Acute kidney injury (AKI) is defined as a sudden loss of the kidney function, resulting in the buildup of waste products, electrolytes and acid–base disturbance as well as inability to maintain fluid balance. Acute decline in kidney function usually follow an injury that causes changes in kidney function or structure [1]. AKI is commonly seen in critically ill patients admitted to intensive care units (ICU), affecting 26–67% of patients [1]. It is associated with poor patient outcomes, such as increased mortality [2], development of chronic kidney disease [3] and end-stage kidney disease (ESKD), increased risk of cardiovascular disease [4] and reduced quality of life [5].

Critically ill patients with AKI are at high risk for dysglycemia [6], which is associated with increased morbidity and mortality. Despite increases in our knowledge on the management of critically ill patients, mortality associated with AKI remains high [7,8].

Tight glycemic control (TGC) has been used in patients at risk of AKI and in the management of those who develop AKI [6]. It has been reported that tight glycemic control can reduce the incidence and severity of AKI [7]. However, as shown in the Intensive Care Evaluation and Survival Using Glucose Algorithm Regulation (NICE-SUGAR), a large randomized multicenter study, survival was increased by conventional glycemic control as opposed to patients managed with TGC [8]. In addition to hyperglycemia, excessive glucose variability is detrimental to critically ill patients [9,10,11] and may be an independent risk factor for adverse ICU composite outcomes [12,13]. Previous studies have shown that factors such as serum creatinine and mechanical ventilation (MV) are independent factors associated with clinical outcomes in patients receiving CRRT [14]. However, the associations between blood glucose levels and glycemic variability and clinical outcomes in severe AKI patients receiving CRRT remain unclear.

The Randomized Evaluation of Normal vs. Augmented Level (RENAL) study is the largest randomized study assessing dialysis dose intensity in patients with severe AKI. We collected data prospectively and conducted a secondary analysis of the RENAL study to explore the effect of baseline blood glucose levels and glycemic variability on the clinical outcomes in patients with severe AKI.

## 2. Methods

The RENAL study was a multicenter, prospective, randomized trial involving 1508 ICU patients with AKI. The detailed inclusion and exclusion criteria along with the study protocol were published in the appendix of the main paper [15]. Briefly, eligible patients were randomized to higher (40 mL/kg/h) versus lower (25 mL/kg/h) intensity of continuous renal replacement therapy. The primary end point was all-cause mortality within 90 days after randomization. The Human Research Ethics Committees of the University of Sydney and all participating sites approved the study protocol, and written informed consent was obtained from all patients.

### 2.1. Blood Glucose Levels

Data on blood glucose levels were obtained at study baseline; the patients were divided into quartiles of baseline BGL.

### 2.2. Glycemic Variability

Standard deviation (SD) and coefficients of variation (CV) were chosen as a measures of glycemic variability [16] (Appendix A). The association between glycemic variability and clinical outcomes was assessed using CV and SD of daily blood glucose level within the 28 days the ICU admission. Patients were then divided into quartiles of CV and SD, respectively.

### 2.3. Study Outcomes

The primary outcome was all-cause mortality at 90 days following randomization. The secondary outcomes were mortality at 28 days following randomization, the duration of hospital stay and length of ICU stay.

The baseline variables that were assessed as outcome predictors included: demographics (age and gender), metabolic and laboratory data (serum potassium [17], creatinine, albumin [18], phosphate [19], bicarbonate [20]) and severity of illness as assessed by mechanical ventilation, sepsis [21] and Acute Physiology and Chronic Health Evaluation III (APACHE III) scores [18] at baseline. The selection of variables was based on identification of all measured clinical variables of known or suspected prognostic importance for the outcomes of interest from the existing literature.

### 2.4. Statistical Analysis

Values were given as mean ± SD for the data with normal distribution, or interquartile ranges for the data with abnormal distribution. Daily glucose variability in continuous variables was presented as CV and SD. The data were compared using Student’s *t*-test or the Mann–Whitney U test, as appropriate. Categorical variables were reported as proportions. The chi-square test was used to conduct a comparison. Cox proportional hazard analysis was used to identify predictors for 90 days’ (28 days’) mortality in the different blood glucose level groups and with different degrees of glucose variability. Analysis of time to death within 90 days of randomization was assessed using the log-rank test and displayed as a Kaplan–Meier curve. Multivariate Cox regression models were performed to assess the association between baseline blood glucose levels and glucose variability with clinical outcomes. The initial analysis was performed without adjustments but was subsequently adjusted for several groups of covariates. Model 1 adjusted for the demographic characteristics, including age and sex. Model 2 adjusted for risk factors associated with mortality, including APS, sepsis, ventilation and other baseline variables, in addition to the variables in Model 1. Model 3 adjusted for variables in Model 2 plus mean glucose. Covariates were selected for inclusion in the model for clinical relevance. Comparisons among different groups were conducted using ANOVA analysis.

In consideration of survival bias, sensitivity analysis was conducted with 7 days’ data. CV and SD were used to correlate blood glucose variability with clinical results. 

All tests were 2-sided, and *p*-values ≤ 0.05 were considered statistically significant. Statistical analysis was performed using SAS version 9.2 (SAS Institute Inc., Cary, CA, USA).

## 3. Results

Glucose levels were available for 1404 out of 1508 patients from the RENAL patients. 1345 out of 1404 patients had data for BG values within 28 days. A total of 627 died out of 1404 patients in the four groups (mortality 51.7% in group 1, BGL < 5.8 mmol/L; 38.7% in group 2, BGL 5.8–7.2 mmol/L; 44.5% in group 3, BGL 7.3–9.1 mmol/L; and 44.0% in group 4, BGL ≥ 9.1 mmol/L, respectively). Baseline characteristics of the study population categorized by the baseline BGL were listed in Table 1. Both lowest and highest BGL groups had higher APACHE III and lower bicarbonate levels than patients of other quartiles. Patients with lower baseline BGL had the greatest mortality (51.7%) and were also more likely to be underweight, with lower albumin levels and lower rates of mechanical ventilation. A total of 472 died out of 1345 patients in the model of glucose variability. Patients in the fourth quartile of glucose variability (CV > 30%, SD > 2.21) had the highest mortality and were more likely to have higher APACHE III scores, lower bicarbonate levels and higher mean glucose levels (Appendix A).

Table 2 shows that more patients in the lowest quartile, with a baseline BGL of <5.8 mmol/L, died at 90 days and 28 days after randomization, compared with patients in the second quartile, who had a near-normal BGL of 5.8–7.2 mmol/L. Crude rates of the primary outcomes were 51.7%, 38.7%, 44.5% and 44% in BGL quartiles 1, 2, 3 and 4, respectively, indicating that patients with near-normal BGL had significantly higher cumulative incidence of all-cause mortality at 90 days (log-rank *p* = 0.02, Figure 1). Multivariate analysis showed that baseline BGL was independently associated with an increased risk of death at 90 days and 28 days (HR 1.48, 95% CI 1.17–1.88, *p* = 0.0013, P trend = 0.02; HR 1.47, 95% CI 1.13–1.92, *p* = 0.04, P trend = 0.09, respectively) after adjustment for priori-defined baseline covariates. Furthermore, there were no significant differences in the duration of hospital stay and ICU stay among quartiles (*p* = 0.55 and *p* = 0.30, respectively; Table 1).

An independent association between glycemic variability and 90-day mortality was also found when multivariate logistic regression analyses were performed to analyze the model of CV and SD. Table 3 shows that more patients in the fourth quartile with CV > 30% died at 90 days after randomization compared with patients in the first quartile (OR 1.40, 95% CI 1.12–1.75, P trend = 0.0002). Multivariate analysis showed that with the increased quartile of CV, HR was increased gradually (P trend < 0.005 for all of the adjusted models) compared with the lowest quartile. The risk of death increased with increasing CV at 90 days (HR 1.02, 95% CI 1.02–1.03, P trend = 0.002; Table 3), with the same results at 28 days (univariate analysis showed OR 1.37, 95% CI 1.07–1.74, P trend = 0.003; multivariate analysis showed HR 1.27, 95% CI 0.97–1.65, P trend = 0.032; Appendix A).

More patients in the fourth quartile (SD > 2.21) died at 90 days after randomization. In the crude model, OR (95% CI) for the highest SD quartile versus the lowest quartile was 1.17 (0.93–1.47) for the primary outcome compared with patients in the first quartile, P trend = 0.082. The mortality at 28 days was almost the same (Appendix A). In Model 1 and Model 2 adjusted for age, sex, APS, sepsis, ventilation and other baseline variables, P trend > 0.05, it was reported that the mean glucose was related to ICU mortality [1]. In Model 3, after being adjusted for Model 2 plus mean glucose, a statistically significant association was found (HR 1.29, 95% CI 1.17–1.42, P trend = 0.027) (Table 4).

With regards to secondary outcomes, hospital stay and ICU stay, CV was significantly different among the four groups (*p* < 0.0001); in the fourth quartile with CV > 30%, the day of hospital stay and ICU stay were significantly shortened, which might be related to the high mortality rate in this group.

### Sensitivity Analysis

More patients in the fourth quartile died compared with those in the first quartile (CV: OR 1.30, 95% CI 1.04–1.63, P trend = 0.0021, 90 days (Appendix A); SD: OR 1.14, 95% CI 0.89–1.46, P trend = 0.33 (Appendix A)). Multivariate analysis showed a gradual increase in HR for all-cause mortality at day 90, with the increased quartile of CV HR 1.02, 95% CI 1.01–1.03, P trend = 0.002 (Appendix A); SD HR 1.26, 95% CI 1.15–1.37, P trend = 0.04 (Appendix A). The relationship between glycemic variability and mortality remained unchanged.

## 4. Discussion

This study was a secondary analysis of the RENAL study and included 1406 participants with severe acute kidney injury receiving continuous renal replacement therapy. The study highlighted the relationship between baseline blood glucose levels, glycemic variability (CV, SD) and poor clinical outcomes, namely risk of mortality and length of ICU and hospital stay. Participants with a baseline blood glucose level within a normal physiological range (BSL < 5.8 mmol/L) were observed to be at higher risk of mortality, although there were no statistically significant differences in the duration of ICU and hospital stay. Increased glycemic variability was also independently associated with poorer clinical outcomes. These findings are particularly pertinent to individuals with severe acute kidney injury receiving continuous renal replacement therapy (CRRT).

The 45% mortality rate reported in this study is consistent with the rates of approximately 50% reported in the literature [21,22]. Our analysis confirms the findings of a large cohort study (n = 66,184) of adult ICU patients in Australia. In this study, early low blood glucose levels (on the day of admission) were also independently associated with an increased risk of hospital mortality [23]. In both crude and adjusted multi-variable analysis, blood glucose levels in the high (8.69–11.79 mmol/L) and highest (>11.79 mmol/L) quartiles were also associated with increased hospital mortality rates. Interestingly, in our study, there was no statistically significant difference in hospital mortality rates between individuals in the high and highest blood glucose quartiles. These disparities could be attributed to differences in our study population, which included individuals with severe AKI requiring CRRT. CRRT is known to affect glucose balance [24] and has the potential to have a significant impact in individuals with severe kidney disease, who, as a population, are already at a higher risk of dysglycemia [6]. In our study, we also found that lower blood glucose levels were associated with greater abnormalities in laboratory parameters (e.g., higher APACHE III scores) and increased severity of illness. Adjusting for risk factors associated with mortality, higher blood glucose levels (BSL > 9.1 mmol/L) tend to benefit patients with severe AKI more so than blood glucose levels in a normal physiological range (BGL < 5.7 mmol/L). While most clinicians agree that glycemic control is desirable in critically ill patients, maintaining optimal blood glucose control remains controversial. Our findings suggest that normal or slightly higher baseline blood glucose levels (BGL > 5.8 mmol/L) produce better clinical outcomes for patients with severe AKI requiring CRRT.

Glucose variability, as opposed to baseline blood glucose levels, is an important factor associated with hospital mortality [16]. Thus, prior observational studies have tended to focus on the relationship between glucose variability and hospital mortality rates in critically ill patients [25]. Glucose variability has been quantified in many different ways, although the most frequently used indicator is the standard deviation [SD]. In our study, more patients died at 90 days in the highest SD quartile when compared to the lowest quartile, although a statistically significant association was only found for adjusted mean glucose levels, which are also independently associated with an increased risk of mortality [6]. In contrast to previous studies where SD data have been extracted from blood glucose levels taken within 24–48 h of admission [26,27], in this study, SD data were obtained from BGL taken in the first 4 weeks of ICU admission. It is well documented in the literature that rapid blood glucose fluctuations can increase oxidative stress, neuronal and mitochondrial damage and coagulation activity [28,29,30]. Chronic blood glucose changes can, however, equally increase oxidative stress and induce chronic inflammation [31]. CV as a preferred measure of glucose variability was deemed to be relatively consistent across the range of mean glucose and HbA1c levels. Although 75% CV was less than 30% in this study, the amplitude of blood glucose fluctuations was still slightly higher than previously reported in the literature [15,32]. The results from our study suggest that CV has a stronger correlation with adverse hospital outcomes. Excluding short-term (admission) variabilities, the study also revealed that fluctuations in glycemic control during hospital stay can significantly impact overall clinical outcomes. The findings emphasize the potential benefit that even small reductions in glycemic variability can have on reducing mortality in patients with AKI receiving CRRT.

Glucose variability occurring at different times during ICU admission may also affect clinical outcomes. Prior studies have tended to focus on glycemic stabilization in individuals with acute kidney injury only during the initial period of ICU admission. In our study, SD and CV data were extracted from daily blood glucose levels measured in the first 4 weeks of ICU admission, allowing for the assessment of changes in blood glucose control over time. This is not only more intuitive to clinicians, but also offers greater clinical utility and may be applied on a daily basis. Further studies are required to explore how long-term glycemic variability may influence the duration of ICU and hospital stay in patients with severe acute kidney injury.

The strengths of this study lie in the large sample size and the long period of follow-up. This is one of the largest studies (n = 1406) investigating the relationship between baseline blood glucose levels, glycemic variability and RRT-related outcomes in patients with severe acute kidney injury receiving CRRT. Known risk factors and interventions independently associated with an increased mortality in individuals with AKI were also considered. This study, however, equally has its limitations. Firstly, this was only an observational study, and although the relationship between BGL, glucose variability and mortality was established, associations do not prove causality. Secondly, it is evident in the literature that blood glucose control influences clinical outcomes differently in critically ill patients with and without diabetes [1]. The diabetic status of participants in our study was not always clearly elucidated, and thus other factors confounding clinical outcomes could not addressed. Finally, our study lacked data for continuous glucose monitoring, and thus the effect of minute glycemic variability could not be observed.

## 5. Conclusions

Blood glucose levels within the normal physiological range at baseline appear to be associated with higher mortality. Glycemic variability is an independent risk factor of clinical outcomes. Normal or slighter high baseline BGL and minimization of glycemic variability may have a significant beneficial impact on clinical outcomes. This effect may be affected by other unmeasured confounds, and it will warrant further studies to prove the effect of blood glucose levels on clinical outcomes in the case of severe AKI.

## Figures and Tables

**Figure 1 jcm-12-00015-f001:**
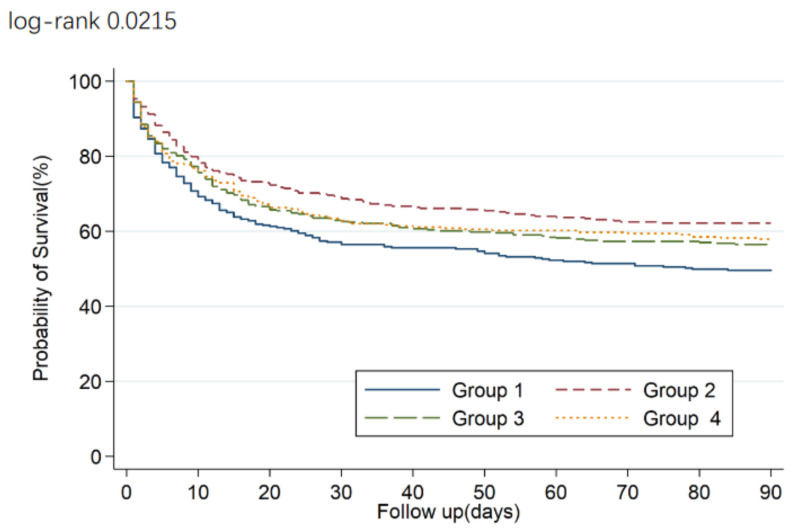
A Kaplan–Meier survival curve.

**Table 1 jcm-12-00015-t001:** Clinical characteristics of baseline glucose level.

Characteristic	Group 1 Glucose < 5.8 (n = 339)	Group 2 5.8 ≤ Glucose < 7.2 (n = 344)	Group 3 7.2 ≤ Glucose < 9.1 (n = 362)	Group 4 Glucose ≥ 9.1 (n = 360)	*p* Value
Age (years)	63.7 ± 15.6	63.7 ± 15.0	64.1 ± 14.5	66.3 ± 14.1	0.06
Male sex–no (%)	190 (56.1)	216 (62.8)	255 (70.4)	241 (66.9)	0.0006
Time in ICU before randomization (hour)	41.5 ± 102.5	58.7 ± 145.9	66.7 ± 140.6	41.3 ± 78.0	0.008
Mechanical ventilation–no. (%)	239 (70.5)	251 (73.0)	275 (76.0)	287 (79.7)	0.03
Severe sepsis–no. (%)	188 (55.6)	169 (49.1)	182 (50.3)	163 (45.3)	0.06
APACHE III score	106.3 ± 25.7	98.2 ± 25.0	99.4 ± 26.2	107.0 ± 24.7	<0.0001
SOFA score	10.7 ± 3.0	10.1 ± 2.9	10.5 ± 2.7	10.2 ± 2.5	0.03
Weight (kg)	78.7 ± 12.7	81.2 ± 12.7	81.7 ± 13.0	80.8 ± 12.9	0.01
Albumin (g/L)	24.9 ± 7.1	26.7 ± 7.2	25.5 ± 6.8	26.8 ± 7.1	0.0004
Creatinine (ummol/L)	341.9 ± 218.4	336.8 ± 197.5	330.1 ± 197.0	323.3 ± 215.0	0.66
Potassium (mmol/L)	4.9 ± 0.9	4.8 ± 0.8	4.8 ± 0.9	4.9 ± 1.0	0.11
Phosphate (mmol/L)	2.05 ± 0.81	2.07 ± 0.83	1.93 ± 0.80	2.02 ± 0.89	0.12
Bicarbonate (mmol/L)	17.4 ± 6.0	19.1 ± 5.9	19.3 ± 5.6	17.7 ± 5.6	<0.0001
No. of days in ICU	12.8 ± 16.7	11.6 ± 13.3	12.6 ± 14.9	10.9 ± 12.2	0.30
No. of days in hospital	25.6 ± 26.2	26.8 ± 24.5	26.8 ± 25.8	24.4 ± 24.4	0.55

**Table 2 jcm-12-00015-t002:** Cox regression analysis on 90-day mortality of baseline glucose level.

Outcomes	Fourth of Glucose		Event, n (%)	Univariate			Multivariate		
	Category	n		HR (95% CI)	*p*-Value	P Trend	HR (95% CI)	*p*-Value	P Trend
Death in 90 days	<5.8	339	175 (51.7)	1.49 (1.19–1.86)	0.0006	0.20	1.48 (1.17–1.88)	0.001	0.02
	5.8–7.2	344	133 (38.7)	1					
	7.2–9.1	362	161 (44.5)	1.21 (0.96–1.52)	0.11		1.22 (0.96–1.55)	0.11	
	>9.1	359	158 (44.0)	1.21 (0.96–1.53)	0.10		1.03 (0.81–1.32)	0.79	

**Table 3 jcm-12-00015-t003:** Cox regression analysis on 90-day mortality of glucose variability (CV).

	Continuous Glucose CV (%)	Quartile of CV (%)				P Trend for Quartile
		Q1 <16	Q2 16–22	Q3 22–30	Q4 >30	
n	1340	356	311	337	337	
Median of Glucose CV	22	12	19	26	37	
Outcomes						
No. of outcomes	568	139	103	157	169	
Crude	1.02 (1.01–1.02)	1	0.78 (0.61–1.01)	1.14 (0.91–1.44)	1.40 (1.12–1.75)	0.0002
Model 1	1.02 (1.01–1.02)	1	0.79 (0.61–1.03)	1.14 (0.90–1.43)	1.37 (1.09–1.72)	0.0007
Model 2	1.02 (1.01–1.03)	1	0.72 (0.55–0.94)	1.05 (0.83–1.33)	1.26 (0.99–1.60)	0.007
Model 3	1.02 (1.02–1.03)	1	0.72 (0.55–0.94)	1.10 (0.86–1.40)	1.37 (1.08–1.75)	0.0008

Model 1 adjusted for age and sex; Model 2 adjusted for age, sex, APS, sepsis (yes or no), ventilation (yes or no) and baseline variables; Model 3 adjusted for age, sex, mean glucose, APS, sepsis (yes or no), ventilation (yes or no) and baseline variables.

**Table 4 jcm-12-00015-t004:** Cox regression analysis on 90-day mortality of glucose variability (SD).

	Continuous Glucose std (mmol/L)	Quartile of STD (%)				P Trend for Quartile
		Q1 <1.03	Q2 1.03–1.52	Q3 1.52–2.21	Q4 >2.21	
n	1340	331	332	342	335	
Median of Glucose std	1.52	0.77	1.26	1.82	2.79	
Outcomes						
No. of outcomes	568	136	125	149	158	
Crude	1.10 (1.03–1.17)	1	0.84 (0.66–1.08)	0.99 (0.79–1.26)	1.17 (0.93–1.47)	0.08
Model 1	1.09 (1.03–1.17)	1	0.85 (0.67–1.09)	0.98 (0.78–1.24)	1.14 (0.90–1.43)	0.15
Model 2	1.27 (1.16–1.38)	1	0.73 (0.56–0.94)	0.86 (0.68–1.10)	1.01 (0.80–1.29)	0.54
Model 3	1.29 (1.17–1.42)	1	0.76 (0.58–0.98)	0.97 (0.76–1.25)	1.33 (1.02–1.74)	0.02

Model 1 adjusted for age and sex; Model 2 adjusted for age, sex, APS, sepsis (yes or no), ventilation (yes or no) and baseline variables; Model 3 adjusted for age, sex, mean glucose, APS, sepsis (yes or no), ventilation (yes or no) and baseline variables.

## Data Availability

Not applicable.

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
