# Peer review of "Prognostic Significance of Baseline Blood Glucose Levels and Glucose Variability in Severe Acute Kidney Injury: A Secondary Analysis from the RENAL Study"

_jcm, 2022, doi:10.3390/jcm12010015_

Round 1
Reviewer 1 Report
This study investigated study the associations between baseline blood glucose levels (BGL), glycaemic variability (GV) and clinical outcomes in patients with severe AKI receiving continuous renal replacement therapy. Further, they identified and concluded that e identified a low BGL within the normal physiological range at baseline and greater CV and SD values as significant modifiable risk factors for mortality in severe AKI patients in ICU, which may be a target of intervention.
Overall, the study is interesting but missing some key points that need to be addressed before final implementation.
Major suggestions: The background section does not provide enough information about AKI. If there is a paragraph which defines AKI under clinical conditions, that would be interesting to see.
Is there any specific information about the predictive value of high glycemic variability on the AKI in patients with young age; hypertension; dyslipidemia; a long duration of diabetes; and who were treated with inhibitors?
How GV differ from HbA1c variability clinically?
Based on these findings, Can you please elaborate more about the possible mechanism of linking GV and AKI?
Is there any adverse clinical outcomes of GV were considered in all patients? Please elaborate.
P2L81: It would be helpful if you rephrased the line so that it is more clear to the reader. Also, the entire MS needs a thorough proofread for typesetting and syntax.
Minor suggestions: P2L77: correct to patients.
P3L100; correct to glucose. P4L141; Correct to multivariate
Author Response
This study investigated study the associations between baseline blood glucose levels (BGL), glycaemic variability (GV) and clinical outcomes in patients with severe AKI receiving continuous renal replacement therapy. Further, they identified and concluded that e identified a low BGL within the normal physiological range at baseline and greater CV and SD values as significant modifiable risk factors for mortality in severe AKI patients in ICU, which may be a target of intervention.
Overall, the study is interesting but missing some key points that need to be addressed before final implementation.
Major suggestions: The background section does not provide enough information about AKI. If there is a paragraph which defines AKI under clinical conditions, that would be interesting to see.
Thanks for the reviewer’s comment. We have added the following paragraph to the background section to provide more information about AKI.
Acute kidney injury (AKI) is defined as a sudden loss of the kidney function, resulting in building up waste products, electrolytes and acid-base disturbance as well as inability to maintain fluid balance. Acute declines in kidney function usually follow an injury that causes changes in kidney function or structure [1]. AKI is commonly seen in critically ill patients admitted to intensive care units (ICU), affecting 26–67% of patients [1]. It is associated with poor patient outcomes, such as increased mortality [2], development of chronic kidney disease [3] and end-stage kidney disease (ESKD), increased risk of cardiovascular disease [4] and reduced quality of life [5].
Is there any specific information about the predictive value of high glycemic variability on the AKI in patients with young age; hypertension; dyslipidemia; a long duration of diabetes; and who were treated with inhibitors?
Thanks for the reviewer’s comment and we totally agreed that it would be interesting to know about the predictive value of high glycemic variability on the AKI in patients with young age and common past medical histories.
Unfortunately, the participants in the RENAL study were not a young population with an average age 64.5 years old. The baseline comorbidities such as history of hypertension; dyslipidemia; a long duration of diabetes; and who were treated with inhibitors were not collected. Therefore we were unable to perform statistical analyses to assess this issue.
How GV differ from HbA1c variability clinically?
Glycaemic variability is fluctuations in blood glucose levels.
There are many indicators that can be used to evaluate glycaemic variability, including standard deviation of blood glucose, coefficient of variation of blood glucose, mean value of the differences in daytime blood glucose and other classic indicators.
Glycated hemoglobin (HbA1c) reflects the average blood glucose levels over the last 2 to 3 months. It is associated with chronic complications. It is unable to reflect glycaemic variability in the acute circumstances.
In our study, we used the blood glucose variability within 28 days to assess its association with clinical outcomes in in this acute ICU population.
Based on these findings, Can you please elaborate more about the possible mechanism of linking GV and AKI?
Blood glucose variability increases oxidative stress, inflammation, neuron and mitochondrial damage, and coagulation activity, which can aggravate kidney injury. The kidney plays an important role in glucose regulation. The sodium-glucose transporter is mainly expressed in the kidney and participates in blood glucose regulation. Acute kidney injury affects glucose metabolism. Furthermore, in patients with severe AKI necessitating CRRT, CRRT also has impacts on blood glucose dynamics and metabolism as well as glucose invariability.
Is there any adverse clinical outcomes of GV were considered in all patients? Please elaborate.
Thanks for the reviewer’s comments.
In this study, we did not collect this information.
In recent years, it has increasingly been recognized
that glycemic variability is a dimension of significant importance among critically-ill patients, independent of the acute highs and lows of blood glucose measurements in the ICU.。
Glycemic variability is an independent predictor of mortality. Current literature suggest that glycemic variability induces oxidative stress, enhances apoptosis, impairs endothelial function, causes mitochondrial damage, and accelerates coagulation activity that can further exacerbate acute renal injury.
P2L81: It would be helpful if you rephrased the line so that it is more clear to the reader. Also, the entire MS needs a thorough proofread for typesetting and syntax.
Minor suggestions: P2L77: correct to patients.
P3L100; correct to glucose. P4L141; Correct to multivariate
Thanks to the reviewer for pointing out this issue.
We have reviewed the paper again and conducted proofreading for typesetting and syntax
Reference
- Hoste, E.A.J.; Kellum, J.A. Acute Kidney Injury: Epidemiology and Diagnostic Criteria. Curr Opin Crit Care2006, 12, 531–537, doi:10.1097/MCC.0b013e3280102af7.
- Zarbock, A.; Kellum, J.A.; Schmidt, C.; Van Aken, H.; Wempe, C.; Pavenstädt, H.; Boanta, A.; Gerß, J.; Meersch, M. Effect of Early vs Delayed Initiation of Renal Replacement Therapy on Mortality in Critically Ill Patients With Acute Kidney Injury: The ELAIN Randomized Clinical Trial. JAMA2016, 315, 2190–2199, doi:10.1001/jama.2016.5828.
- Heung, M.; Steffick, D.E.; Zivin, K.; Gillespie, B.W.; Banerjee, T.; Hsu, C.-Y.; Powe, N.R.; Pavkov, M.E.; Williams, D.E.; Saran, R.; et al. Acute Kidney Injury Recovery Pattern and Subsequent Risk of CKD: An Analysis of Veterans Health Administration Data. Am J Kidney Dis2016, 67, 742–752, doi:10.1053/j.ajkd.2015.10.019.
- Legrand, M.; Rossignol, P. Cardiovascular Consequences of Acute Kidney Injury. N Engl J Med2020, 382, 2238–2247, doi:10.1056/NEJMra1916393.
- Vijayan, A.; Abdel-Rahman, E.M.; Liu, K.D.; Goldstein, S.L.; Agarwal, A.; Okusa, M.D.; Cerda, J.; AKI!NOW Steering Committee Recovery after Critical Illness and Acute Kidney Injury. Clin J Am Soc Nephrol2021, 16, 1601–1609, doi:10.2215/CJN.19601220.

Reviewer 2 Report
I think this paper is a useful study of whether baseline blood glucose affects prognosis. And you clearly showed that the prognosis is worse with lower blood glucose (while still in the normal range) and with greater variability in blood glucose. I believe this paper is an appropriate paper for this journal.
I have some opinions, so please consider them.
Major Revision
You have shown that the prognosis is worse with lower blood glucose, but I think it is important to clarify that we do not know in this study whether that result is a cause or a consequence. In other words, if it is cause, the prognosis will improve with strict control of blood glucose, but if it is a result, it is not likely to improve. You wrote a little bit at the end.
Statistical method: For the studies in Table 1 and Table 3, you are comparing multiple groups. The method is not mentioned in the statistical methods section; I assume it is ANOVA or Kruskal-Wallis test. Please clarify.
This study is a sub-study and the original study was a prognostic comparison with higher and lower intensity of continuous renal replacement therapy. I would be interested to know what was done about glucose in those two groups.
Minor Revision
Hyphenated words in the middle of sentences in some places are not necessary (e.g. line 46: sever-ity → severity etc.).
Author Response
review2
I think this paper is a useful study of whether baseline blood glucose affects prognosis. And you clearly showed that the prognosis is worse with lower blood glucose (while still in the normal range) and with greater variability in blood glucose. I believe this paper is an appropriate paper for this journal.
I have some opinions, so please consider them.
Major Revision
You have shown that the prognosis is worse with lower blood glucose, but I think it is important to clarify that we do not know in this study whether that result is a cause or a consequence. In other words, if it is cause, the prognosis will improve with strict control of blood glucose, but if it is a result, it is not likely to improve. You wrote a little bit at the end.
Thanks for the reviewer’s comment.
We totally agreed with you that the causal relationship between low blood glucose levels and prognosis. In fact, this clarification was stated in the discussion section under the first limitation.
Statistical method: For the studies in Table 1 and Table 3, you are comparing multiple groups. The method is not mentioned in the statistical methods section; I assume it is ANOVA or Kruskal-Wallis test. Please clarify.
Thanks for the reviewer’s comment.
Yes we used ANOVA analysis to compare multiple groups listed in the table 1 and 3. We have added this information in our manuscript.
This study is a sub-study and the original study was a prognostic comparison with higher and lower intensity of continuous renal replacement therapy. I would be interested to know what was done about glucose in those two groups.
Thanks for the reviewer’s comment.
We agreed that intensity of CRRT may have an impact on glucose metabolism and variability. Hence, we are planning to assess this issue as a separate project.
Minor Revision
Hyphenated words in the middle of sentences in some places are not necessary (e.g. line 46: sever-ity → severity etc.).
Thanks to the reviewer for pointing out this issue.
In our original manuscript that was submitted the journal, there were no such unnecessary hyphenated signals. We were uncertain if it was related to formatting during the process of the manuscript submission. Nonetheless, we have reviewed the paper again and removed all unnecessary hyphenated signals. We have also consulted the journal associated editor regarding this formatting issue.
